# Influence of Pre-Milling of Cr_3_C_2_-25 NiCr Spray Powder on the Fatigue Life of HVOF-Sprayed Coating on ASTM A516 Steel Substrate

**DOI:** 10.3390/ma16041593

**Published:** 2023-02-14

**Authors:** Rosivânia da P. S. Oliveira, Gabriel R. Cogo, Brenno L. Nascimento, Matheus M. S. Reis, Antonio Takimi, Sandro Griza, Carlos P. Bergmann

**Affiliations:** 1Post-Graduation Program in Materials Science and Engineering, Federal University of Sergipe, São Cristóvão City 49100-000, SE, Brazil; 2Post-Graduation Program in Mining, Metallurgical and Materials Engineering, Engineering School, Federal University of Rio Grande do Sul, Porto Alegre City 90035-190, RS, Brazil; 3Rijeza Metallurgical Industry, São Leopoldo City 93140-000, RS, Brazil

**Keywords:** Cr_3_C_2_-25NiCr, spray powder, high-energy milling, HVOF, fatigue

## Abstract

The aim of the present investigation is to evaluate the influence of the powder size of Cr_3_C_2_-25NiCr spraying powder on the fatigue behavior of HVOF-sprayed coating on the ASTM A516 steel substrate. Conventional commercial Cr_3_C_2_-25NiCr spraying powder was previously treated through high-energy milling. The crystallite sizes of milled powders were measured by X-ray diffraction and transmission electronic microscopy. Three different powder formats of the same Cr_3_C_2_-25NiCr composite were subjected to HVOF spraying to produce (i) a Milled-Coating (from high-energy milled spray powder), (ii) an Original-Coating (from conventional commercial spray powder), and (iii) a 50%–50% mixture of both (Milled + Original-Coating). The same spraying conditions were adopted for all the assessed cases. The sprayed coatings were investigated through the Knoop hardness test and SEM-EDS analysis. In addition, 3-point bending fatigue tests were conducted at different stress levels up to 10^7^ cycles. The coating morphology and roughness effects on fatigue behavior were analyzed. The Cr_3_C_2_-25NiCr milled coating presented a lower fatigue life above the fatigue limit and a higher fatigue limit than other coatings; this outcome could be attributed to its lower surface roughness and finer grain size microstructure.

## 1. Introduction

Protective coatings can increase the lifespan of equipment parts and structures, since the coating protects the substrates from damaging mechanisms, such as corrosion, fatigue, high-temperature oxidation, and wear [1,2,3,4,5,6].

Fatigue failure is a concerning phenomenon since the propagated cracks can reach a critical size and lead to a catastrophic rupture of the component [7]. The roughness of surfaces subjected to positive stress in fatigue loading processes is a relevant parameter since fatigue nucleation depends on stress raisers attributed to the surface roughness. Furthermore, residual stress is considered a critical factor for fatigue behavior [8]. Fatigue performance is improved by surface compressive stress, which hinders crack nucleation since the compressive stress is additive to the external load used to determine the effective cyclic stress acting in the component [9]. One of the prominent benefits provided by coatings lies in their potential to promote compressive residual stresses on the coating surface [7]. Some operating parameters, such as feedstock features, deposition rates, and differences in the coating and substrate’s thermal expansion coefficient, have shown significant influence on coating strain and stress distribution [8].

Nowadays, metal alloys and composites applied as a protective coating in equipment that works in aggressive environments has been a subject of increasing interest [10,11,12,13]. The conception of some coatings is based on NiCr alloy as a metal matrix, providing toughness to the coating, as well as improving the corrosion and oxidation behavior. Dispersed in the NiCr matrix are chromium carbide particles (Cr_3_C_2_), which are well known and widely applied for wear protection purposes, promote mechanical strength, and act against erosive and abrasive wear [6,9,10]. Cr_3_C_2_ presents higher strength, coating adhesion, low density, and good chemical stability [9].

Nanostructured coatings have the potential to enhance physical and mechanical properties, such as mechanical strength, hardness, ductility, specific heat, and diffusivity [14]. Coatings resulting from powders milled down to the nanometer level, whose structure includes nanometric crystallites, provide higher adhesion to the substrate, significantly lower porosity, and higher surface quality, in addition to increased hardness and wear behavior [9]. High-energy milling (HEM) is the technique whose powder mixes of different materials are ground together to obtain homogeneous alloys [15]. This process causes intense plastic strain in powders, which leads to the hardening of the particles, fracture, and, subsequently, to cold welding at successive stages [16]. HEM can promote the fracture of the particles up to the nanometer scale. Furthermore, the structure of the particles undergoes intense dislocation multiplications; this process reduces the long-range order in the particles and leads to crystallite formation. Moreover, the main objectives of using HEM to synthesize Cr_3_C_2_-25NiCr coating powder are: (i) reducing composite granulometry; (ii) ensuring that the carbide particles are surrounded by the precursor constituent of the NiCr binder phase; and (iii) creating new active surfaces and structural defects in the composite. Reducing the particle size of the powder increases the grain boundaries of the coating’s microstructure, which can increase the layer hardness due to dislocation-related restrictions [17].

The coating method can also play an important role in the resulting properties of the coating layer [17]. High velocity oxygen fuel (HVOF) thermal spraying is a technique based on the hypersonic flame at speeds up to 3000 m/s that releases high energy, which is converted into heat and pressure [3,17,18,19,20,21]. Particles reach the substrate and form a dense and hard coating, with low porosity and high adhesion strength of the coating to the substrate after the HVOF deposition [8,22,23,24]. The HVOF hamper phase transformations due to the faster powder speed, heating and cooling, and the lower heat transfer to the feedstock. HVOF-sprayed nanostructured materials can provide coatings with more strength to crack propagation [14]. Although conventional thermal spraying makes particles melt, HVOF’s temperature and spraying speed cause plastic strain in the powder upon impact against the substrate and form mechanically bonded flattened portions of the deposited material, sometimes called splats, that cool at rates up to 1 × 10^6^ K/s [4].

Coatings deriving from nanostructured powders have been recently investigated, and they appear to enhance coating performance [1,6,9,25]. It is well established that coatings increase the wear and corrosion performance [3,9,26,27]; however, the coatings’ fatigue performance plays an important role in equipment whose operations are conducted under cyclic stress. Several concerns about coating fatigue [8,14,28] highlight the need for broadening the knowledge of this matter. The literature still lacks studies focused on investigating fatigue in Cr_3_C_2_-NiCr coatings obtained from HEM. Therefore, it is important to evaluate the fatigue strength of the coatings manufactured from milled powder, as well as from powder blends containing 50%, to determine if it is advantageous to expend the highest amount of energy in milling to achieve a fatigue strength advantage.

The aim of the present study was to evaluate the fatigue performance of coatings consisting of particles of Cr_3_C_2_ and NiCr matrix, both in the milled and original format, and the performance of the 50–50% combination of both, sprayed on an ASTM A516 steel substrate by HVOF technique.

## 2. Materials and Methods

### 2.1. Spraying Powder Milling and Characterization

Conventional commercial Cr_3_C_2_-25NiCr spray powder (PRAXAIR Surface Technologies) containing 75 wt% Cr_3_C_2_, 20% Ni, and 5% Cr was used in the present study. The average powder size was 45 μm based on the manufacturer’s specifications and by an overview SEM image with low magnifications of commercial Cr_3_C_2_-25NiCr powders [29].

The Pulverisette 6 planetary mill, equipped with a steel-coated grinding jar and 5 mm zirconia balls as the grinding media, was used to evaluate the effectiveness of the HEM method. The mass spheres/raw material ratio was 5:1. The adopted milling speed was 800 rpm; 20 mL of ethyl alcohol and 3 wt% of alumina were used as the milling medium to increase the grinding efficiency [30]. After milling for 12 h, the powder was oven dried at 60 °C for 20 min and sieved to 325 # (45 µm) according to the recommended protocol [17].

X-ray diffraction (XRD) analysis enabled the identification of the powder’s phases after milling. This procedure was conducted in a Philips X-ray Analytical Equipment X’Pert-MPD System, equipped with a PW3040/00 and PW3373/00 console, as well as with CuKα anode. The herein adopted parameters comprised 40 kV and 40 mA, 10 mm window, 1° slot, and an angle between 5° and 75°. The diffractograms were treated and analyzed in WinFit 1.2 software; the single-line method was used to calculate the mean crystallite size [15]. Scanning electron microscopy (SEM TESCAN VEGA LMS) and energy dispersive spectroscopy (EDS JOEL Carry Scope JSM-5700) techniques were applied for powder and coating analysis purposes.

Transmission electron microscopy analyses (TEM) were carried out in JEOL JEM 1440 plus equipment at 120 KV. The samples were dispersed in isopropyl alcohol and subjected to an ultrasound bath for 8 min. An aliquot of 5.00 μL of this dispersion was used on a copper grill coated with carbon film for analysis purposes.

### 2.2. Spraying Process and Coating Characterization

Prior to HVOF-spraying, the substrates of ASTM A516 steel, measuring 75 × 15 × 5 mm^3^ (Figure 1), were machined and blasted on the surface by alumina powder to enhance the coating anchoring during the HVOF spraying.

The Sulzer Metco DJ2700 model was used as the spraying equipment. The HVOF-spraying parameters are listed in Table 1. Three different coatings were produced: (1) a coating made from high-energy milled powder (Milled-Coating), (2) a coating made from commercially available powder (Original-Coating), and (3) a coating consisting of 50% of a mixture of both (Milled + Original-Coating).

Metallographic samples were prepared from the sprayed coatings for microstructure analysis, coating thickness, porosity, and microhardness measurements. The samples were sanded up to 2000 # and polished with a 1 μm diamond paste. Three metallographic samples of each coating were prepared. Six images of each sample were obtained at different coating sites and treated in Axiovision software 4.9.2 for porosity analysis purposes when the gray level amplitudes were sufficient to identify the pores [31].

A Buehler Micromet 2001 microhardness tester was used to apply the Knoop microhardness (HK) test to the coatings. The Knoop scale was selected to ensure the adequate microhardness measurement of the thin layers. Five indentations per metallographic sample were performed by applying a load of 0.05 kgf [32]. The maximum depth of indentation of the coatings was 0.77 microns.

The roughness (Ra) was longitudinally evaluated in 5 specimens for each group after spraying. The Mitutoyo Roughness SJ-410 was applied at 0.2 mm/s and at a median line of 2.5 mm to give 10 measurements of each specimen.

The 3-point bending fatigue test was performed by MTS Landmark 370.02 to give the S-N curve of each coating group. The tests were evaluated according to ASTM E466-07 and ISO 22407:2021 standards. The applied parameters were 30 Hz frequency, a loading ratio R of 0.1, a limit of 5 × 10^6^ cycles, cylinders of 5 mm diameter as the supports and plunger, and a 30-mm spam. At least two specimens were tested for each maximum cyclic stress level between 560 MPa and 340 MPa. Figure 1 shows the configuration of the specimen in the 3-point bending fatigue apparatus, where the spacing between the supports was fixed at 30 mm to achieve bending and to avoid shear stresses. Figure 1a shows the schematic of the specimen and the position of the coating in the fatigue test apparatus, whereas Figure 1b exhibits the specimen and its positioning in the fatigue equipment.

## 3. Results and Discussions

### HEM Crystallite Size Reduction

Figure 2 explains the XRD analysis applied to the original Cr_3_C_2_-25NiCr spraying powder as a function of high-energy milling time, indicating the formation of Cr_3_C_2_ and Cr_7_C_3_ during the HEM process. This formation has been reported in the literature by other authors [15,17,21,30,31,33].

The intensity of the peaks decreased, but their width increased as a function of the milling time. After 12-h, the Cr_3_C_2_ and Cr_7_C_3_ peaks persisted, suggesting that the crystalline structure has been preserved to some extent, although the reduction of the crystallite size by atom rearrangement is due to high strain energy. Furthermore, the Cr_2_O_3_ peak emerged as the HEM time grew longer (at least, 5 h), which can be explained by the increased amount of the Cr_2_O_3_ protective film surrounding the increased number of free surfaces of new fine particles. Figure 2 also denotes the disappearance of the Cr_7_C_3_ phase with an increase in the Cr_2_O_3_ phase and milling time. XRD analysis of the NiCr matrix phase transformations was not performed because the major concern of this research was on carbides, and the significant phase transformation of the metal matrix by plastic deformation was not expected.

X-ray diffraction analyses of the original and milled powders were used to determine the crystallite size, the microdeformation of the crystalline lattice, and the lattice parameter of a solid solution based on NiCr. In the early stages of HEM, the size of the original crystallite decreased rapidly from 140 nm to 50 nm. This calculation was made using the Cr_3_C_2_ peak near 39° of diffraction [34]. Figure 3 explains the development of the crystallite sizes of the Cr_3_C_2_-25NiCr spray powder as a function of the high-energy milling time. The values were obtained by means of the XRD technique and peak broadening measurements.

The HEM process significantly reduced the crystallite size of the original Cr_3_C_2_-25NiCr spraying powder after the first two hours. The reduction tended to be saturated after 5 h of milling. After 12 h of milling, the crystallite size of the Cr_3_C_2_-25NiCr spraying powder reached 13 nm.

The surface morphology of the milled powder was analyzed using SEM and TEM images (Figure 4). The agglomeration of the fragments of the original particles (Figure 4a,b) may occur after 12 h of grinding due to their high surface energy and electrostatic strengths between the particles. The TEM Images (Figure 4c) show the milled particles by highlighting the Cr_3_C_2_-25NiCr composites with sizes smaller than 100 nm and irregular shapes.

Figure 5 shows the SEM images of the original and milled Cr_3_C_2_-25NiCr particles. Figure 5a explains the highly porous morphology normally found in the conventional original powder particles [35], with the spherical shape attributed to atomization to increase the flow index of the powder [7]. Figure 5b shows the strong agglomeration of the milled particles due to the reactivity between the particles according to HEM. In the practical procedure, the powder flow was significantly affected by the agglomeration of the particles when we introduced the milled powder for spraying, which requires the use of a dispersant mixed with the milled particles when this material is used over a large area.

Figure 6 exhibits the SEM image of the mixed powder group. The original particles are surrounded by the milled particles. 

Figure 7 denotes the representative SEM metallographic images of the coating with powder (a) milled, (b) original, and (c) 50% mixed. The dense microstructures are typical of carbide coatings sprayed with high-velocity flaming [35]. The light gray phase is the NiCr matrix distributed between the carbide particles (dark gray phase). The darker areas were also observed, which are the characteristic pores of the coatings [14].

In Figure 7a, the laminar architecture of the layer can be observed, which has a splat morphology of different natures. The lightest gray phase of the layer observed in the SEM analysis refers to the NiCr metal, while the darkest phase indicates the Cr-rich Cr_3_C_2_. The carbide particles are larger in the original layer (Figure 7b) than the carbides in the milled layer (Figure 7a). In the 50% mixed layer (Figure 7c), the carbides are present both as platelets together with the matrix and as equiaxed particles, indicating the heterogeneity of the microstructure of the layer.

The parameters of the spraying process and the composition of the powder were the same for all the coatings. The heterogeneity of particle size may affect the formation of deposits, especially for particles with high hardness. The short residence time of the particles in the HVOF flame and the low heat conduction to the inner part of the agglomerate mean that larger particles do not reach thermal equilibrium in flight. Therefore, its deformability may be limited when it hits the substrate, which could explain the microstructural heterogeneity of the 50% mixture layer. However, this event does not occur in the other two layers, where the splat structure is more homogeneous. Moreover, the milled layer exhibits the microstructure with more refined splats and carbides [33].

The microhardness and thickness of the milled, original, and 50% mixed coatings were statistically the same, as shown in Table 2. The porosity of the coatings did not undergo significant differences for all types of coatings. These parameters are often affected by the type of thermal spray and the composition of the coating; however, the parameters studied here did not change. However, the surface roughness of the coatings exhibited differences. The roughness results from the deformation (flattening) and compression processes that are observed during the successive impacts of the sprayed particles on the substrate surface, as well as from the particle size distribution, which can often have a large influence on the surface roughness [33,35].

The roughness is significantly influenced by the type of coating. The coated surface of a given component is usually machined after the coating process to reduce the roughness. However, the machining process was not the subject of this study. Instead, we wanted to analyze the effects of the roughness on the fatigue behavior of the coatings. The surface of the coatings is shown in Figure 8. The images denote the correlation of the surface features with the previously evaluated microstructures and also adequately reflect the measured roughness results. The carbides are refined and more evenly distributed in the milled coating (Figure 8a). The carbides are coarser in the original coating (Figure 8b), while the heterogeneity of the particles can be seen in the 50% mixed coating (Figure 8c).

Figure 9 exhibits the S-N diagram with the curves of the milled, the original, and the 50% mixed coating. The value of 550 MPa was used to start the fatigue tests because this is the yield strength of the ASTM A516 Grade 60 carbon steel substrate. Thus, the highest level of the maximum cyclic fatigue stress was the yield strength. All the other levels were below the yield strength, in the elastic range of the steel, since this is the range of several mechanical designs, where the phenomenon of fatigue fractures due to cyclic loading can occur. As the stresses were reduced in the fatigue tests, the graph exhibits an increase in the average scatter of the fracture points of the specimens, indicating that at each stress level tested, regardless of the coating, the distance points are a natural behavior of ASTM A 516 carbon steel.

From the inclined line of the diagram, we can see that milled and the 50% mixed coatings contribute with essentially the same fatigue strength, as their sloped lines converge, indicating that their specimens fractured with a very similar number of cycles. This may indicate that HEM reduces the performance of the coating at the finite life of the specimen against fatigue, even with the 50% mixed coating, as there was a prevalence of the milled.

From the horizontal line in the graph, we can conclude that the milled coating contributes positively to the fatigue limit of the specimen. Oscillations below 380 MPa reached infinite life (the specimens did not fracture) only for the specimens coated with the milled powder, while the original and the 50% mixed specimen reached the fatigue limit only below the stress of 340 MPa.

The fatigue limit reached 380 MPa for the milled coating, which tends to be 40 MPa higher than for the other coatings (Figure 9). The development of fatigue cracks can be accelerated by increasing the roughness [19]. Milled coatings tend to have a higher fatigue limit due to their lower roughness, which increases the critical stress for fatigue nucleation. In addition, the refined microstructure of the milled coating may be associated with the better fatigue behavior.

Figure 10 exhibits the SEM representative image of all specimens fractured during fatigue testing. The ratchet marks highlight several fatigue nucleation sites on the coated surface (Figure 10a). Crack propagation can be seen on the smooth surface that precedes the final fracture—with a rougher aspect. Figure 10b shows the fatigue striations found as fatigue propagates through the substrate.

The SEM images of the milled (Figure 11a), original (Figure 11b), and 50% mixed (Figure 11c) coatings explain that the fragmented appearance of the fractured surfaces without static micromechanisms—such as dimples, cleavage, or intergranular separation—as well as the appearance of some striations indicate the fatigue process of the coatings. The more refined appearance of the milled coating fracture is in agreement with the findings of the microstructural analysis and the roughness analysis of the groups.

Fatigue nucleated on the surface of the coatings, as exhibited in Figure 11, denotes that there are fatigue striations on the coating coming from the surface. The limit between the fracture plane and the coating surface is indicated in Figure 11b. Fatigue processes began at the surface of the layers and propagated across the interface with the substrate and continued to form striations on the substrate. Since the roughness and microstructure of the milled layer are more refined, this layer experienced the better fatigue performance.

## 4. Conclusions

The objective of the present study was to evaluate the fatigue performance of coatings consisting of Cr_3_C_2_-NiCr composites of (i) conventional commercial original spray powder; (ii) conventional commercial spray powder subjected to a high-energy milling process (HEM); and (iii) a 50–50% combination of both powder types sprayed onto an ASTM A516 steel substrate using the HVOF technique. The investigation carried out here led to the following conclusions:
The milled coating presented lower roughness and more refined microstructure than the original and mixed coatings;The milled coatings exhibited a lower fatigue life above the fatigue limit and a higher fatigue limit than the original and mixed coatings;The coatings’ thickness and hardness did not denote significant differences among the coating types after applying the parameters used in the study.

## Figures and Tables

**Figure 1 materials-16-01593-f001:**
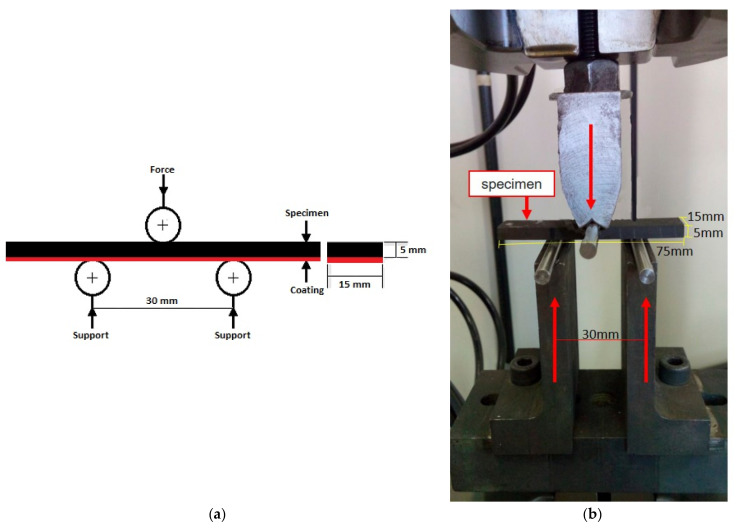
Specimen configuration for the 3-point bending fatigue test. (**a**) schematic of the specimen and the position of the coating in the fatigue test apparatus, (**b**) the specimen and its positioning in the fatigue equipment.

**Figure 2 materials-16-01593-f002:**
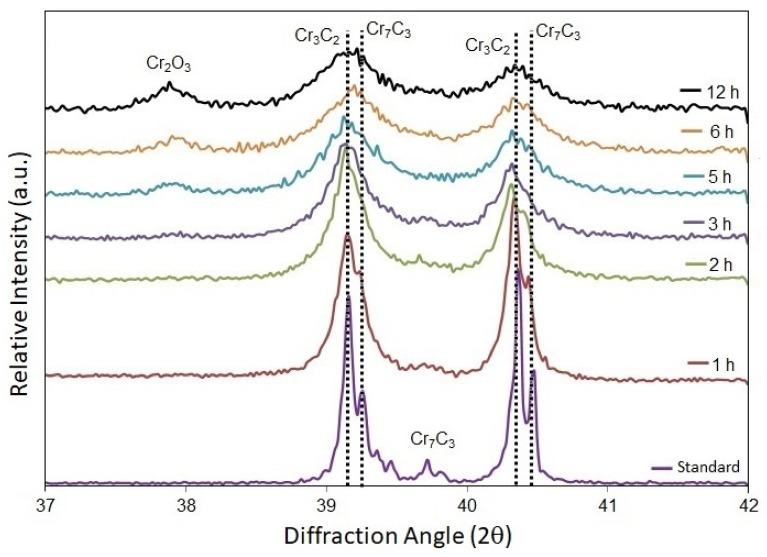
XRD analysis of the Cr_3_C_2_ and Cr_7_C_3_ as a function of high-energy milling time.

**Figure 3 materials-16-01593-f003:**
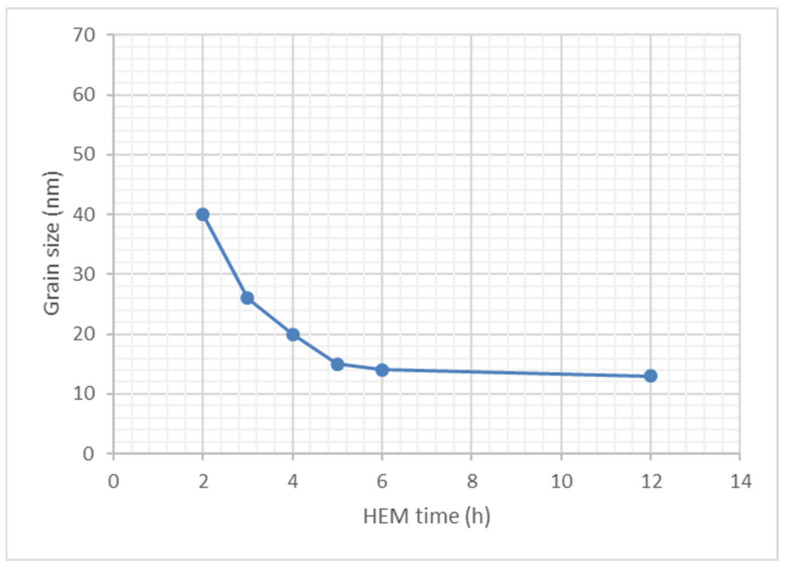
Grain size as a function of high-energy milling time. There was a rapid decrease in the size before the 5-h milling, reaching close to the saturation value of the size reduction.

**Figure 4 materials-16-01593-f004:**
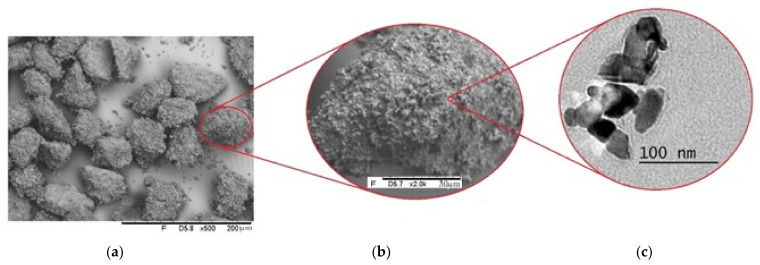
SEM images of the Cr_3_C_2_-25NiCr milled particles. (**a**) Sets of milled particles agglomeration. (**b**) Agglomerated milled particles in more detail. (**c**) TEM images showing the irregular shape of single-milled particles.

**Figure 5 materials-16-01593-f005:**
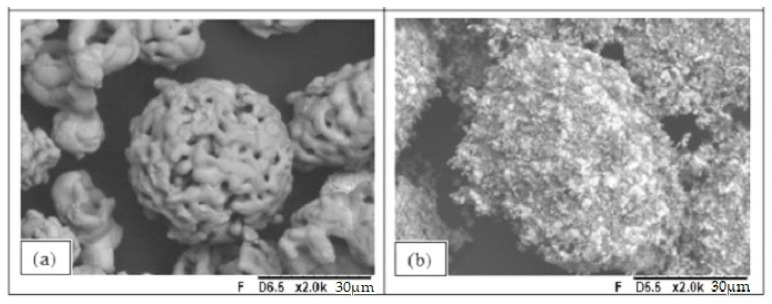
SEM images of (**a**) the original spraying powder and (**b**) after HEM spraying powders.

**Figure 6 materials-16-01593-f006:**
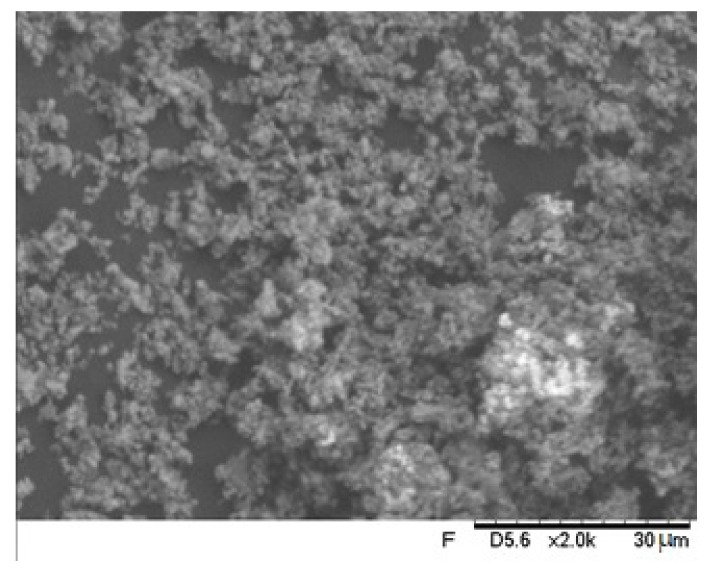
SEM images of the 50% mixed spray powder. The arrow denotes a gross fragment of the original particle surrounded by the milled particles.

**Figure 7 materials-16-01593-f007:**
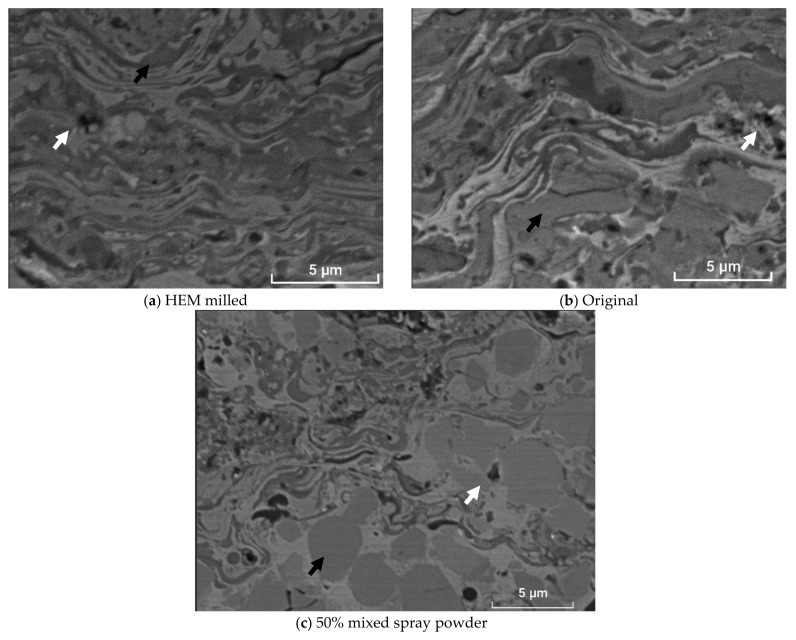
SEM images of the coating with powder (**a**) milled, (**b**) original, and (**c**) 50% mixed. The dark gray phases are Cr-rich Cr_3_C_2_ regions, while the light gray phases represent the NiCr matrix. The dark arrows indicate carbides surrounded by the matrix. The white arrows indicate some porosities.

**Figure 8 materials-16-01593-f008:**
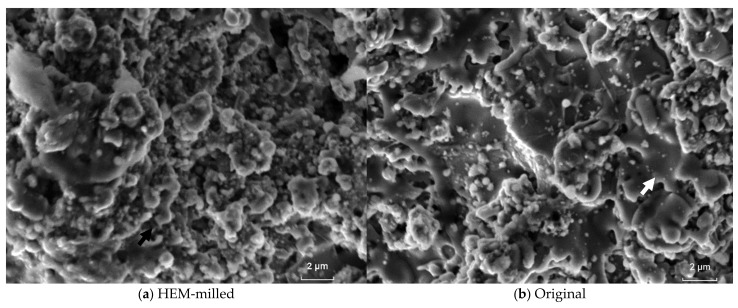
SEM images of the coating surface made from (**a**) HEM-milled spray powder, (**b**) original, and (**c**) 50% mixed spray powder. The surfaces are without any machining steps. The white arrow denotes carbide splats.

**Figure 9 materials-16-01593-f009:**
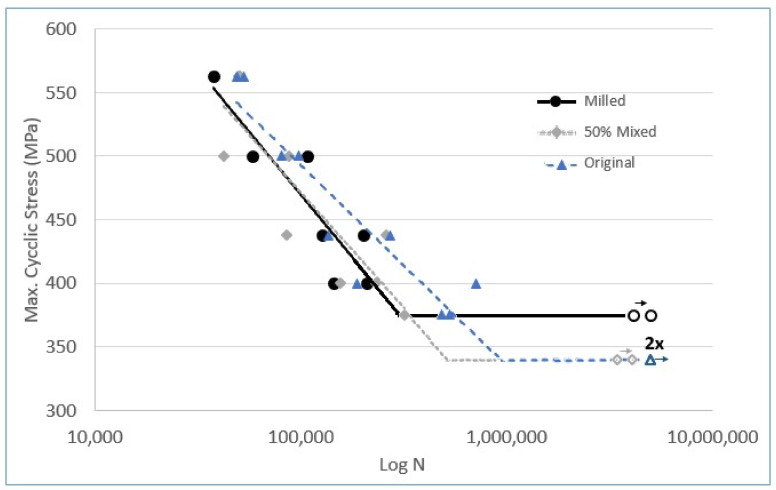
S-N curves of the coating produced from HEM-milled, original, and 50% mixed spray powder. The open dots represent specimens that did not fracture during the fatigue test.

**Figure 10 materials-16-01593-f010:**
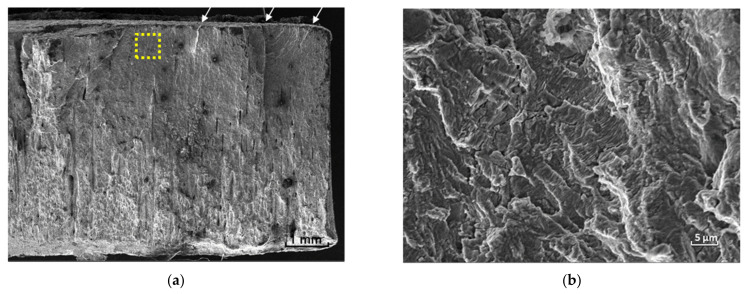
(**a**) Representative SEM image of the coated specimen fracture. The ratchet marks (some of them are highlighted by arrows) at the top of figure indicate fatigue nucleation sites. Crack propagation is identified by the smooth surface preceding the final fracture. (**b**) Fatigue striations perceived on the region highlighted in (**a**).

**Figure 11 materials-16-01593-f011:**
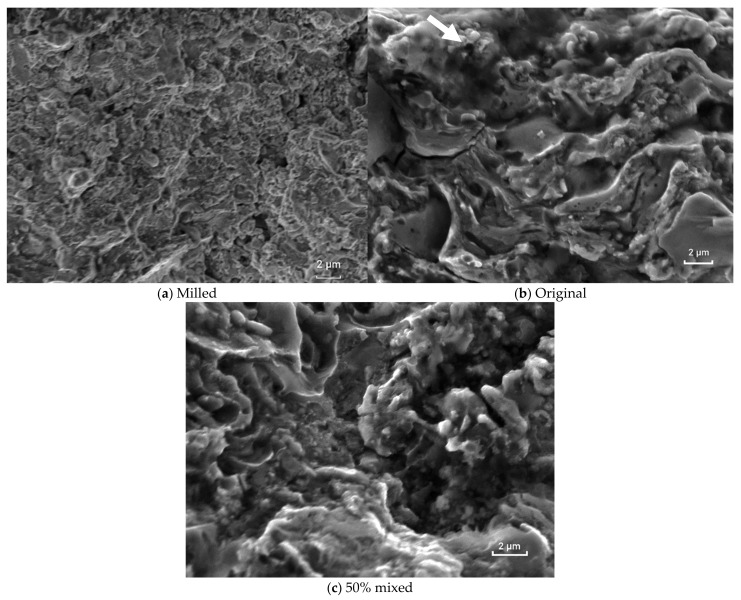
SEM images of the fractured surfaces of the coatings made from (**a**) HEM-milled; (**b**) original—the arrow denotes the outer surface of the specimen; and (**c**) 50% mixed spray powder. Fatigue striations can be seen coming from the coating surface.

**Table 1 materials-16-01593-t001:** HVOF spraying parameters of the Cr_3_C_2_-25NiCr coating.

Parameters	Value
Oxygen pressure (PSI)	150
Oxygen flow (ft^3^/h)	560
Propane pressure (PSI)	100
Propane flow (ft^3^/h)	160
Air pressure (PSI)	100
Air flow (ft^3^/h)	850
Feeding rate (g/min)	35
Spraying distance (mm)	230
Number of passes	01
Spraying angle (°)	90

**Table 2 materials-16-01593-t002:** The coatings’ properties. Standard deviations are shown in parenthesis.

Feature	Coating		
	Milled	Original	50% Mixed
Hardness (HK)	1166 (189)	1081 (0)	1177 (232)
Thickness (μm)	42 (9)	40 (8)	50 (5)
Roughness Ra (μm)	1.27 (0.03)	1.41 (0.05)	1.58 (0.04)

## Data Availability

Not applicable.

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
