# Peer review of "Influence of Pre-Milling of Cr3C2-25 NiCr Spray Powder on the Fatigue Life of HVOF-Sprayed Coating on ASTM A516 Steel Substrate"

_materials, 2023, doi:10.3390/ma16041593_

Round 1

Author Response

Aracaju, January 30th 2023

Mr. Professor Reviewer 1

RE: Article submission

This is the revised version of the article entitled “Influence of grain size of Cr3C2-25 NiCr spray powder on the fatigue life of HVOF-sprayed coating on ASTM A516 steel substrate”. Thank you very much for allowing us to submit the revised manuscript. We would like to thank the your recommendations. We carefully analyzed all comments, as follows. All changes made into the manuscript are marked in yellow.

Reviewer 1.:

As per reviewer comment 1: “The manuscript presents a work on the effect of grain size of powders on the fatigue life of HVOF-sprayed Cr3C2-25NiCr coatings, which is of interest to scientific community. However, without detracting from it, the work is not deep enough in its depth of analysis and the interpretation and justification of results. And the discussion, especially the influence of grain size on the fatigue life, is not clearly addressed. As stated in the manuscript, the lower surface roughness would also attribute to higher fatigue life, but this outcome is more based on references to previous literature rather than robust and detailed experimental findings from this work. While the experimental data by itself is useful as a basis for comparison for more in-depth future works, this alone may not be published in my opinion.”

Response: Thank you for these comments. We have tried to add a little more depth to the discussions on the influence of roughness and microstructure on fatigue strength of the coatings.

Question 1. Page 1 Line 43, the first sentence. Is nickel-chromium (NiCr) alloy or Cr3C2-25NiCr applied as protective coatings, please check since the whole manuscript is about Cr3C2-25NiCr.

Response: Thank you for this comment. We changed the text accordingly: “Nowadays, metal alloys and composites applied as protective coating in equipment that works in aggressive environments has been a subject of increasing interest [10, 13]. The conception of this coating is based on NiCr alloy as metal matrix, providing toughness to the coating, as well as improving the corrosion and oxidation resistance”.

Question 2. Page 2 Line 86. It is suggested that “nanostructured Cr3C2 and NiCr coatings” should be revised as Cr3C2-NiCr coatings.

Response: Thank you for this comment. We changed the text accordingly: “The literature still lacks studies focused on investigating fatigue in Cr3C2-NiCr coatings obtained from HEM.”

Question 3. Page 3 Line 98. It is suggested to provide an overview SEM image with low magnifications of commercial Cr3C2-25NiCr powders. Providing a detailed morphology of a single Cr3C2-25NiCr particle with hard particles (Cr3C2) and binder phases (NiCr) marked respectively, would be better.

Response: We agree with this recomendation, and we added the following sentence to the paper: “The average powder size was 45 μm, based on the manufacturer's specifications, and by an overview SEM image with low magnifications of commercial Cr3C2-25NiCr powders, as we will explain later.”

Question 4.  Fig. 5. The scale bar is not clear.

Response: Corrected. The scale bar has been changed.

Question 5. Fig. 8, it is difficult to tell the difference between the carbide distribution on three kinds of coatings. Please mark the carbide particles.

Response: In Fig. 8 some carbide particles have been marked by arrows.

Question 6. As the author stated, the better fatigue behaviour might be associated with the refined microstructure. However, the discussion in not presented in detailed.

Response: We would like to thanks the reviewer for this recommendation. We changed the discussion as follows: “Figure 9 shows a SN diagram with the curves of the milled, the original and the 50% mixed coating. The value of 550 MPa was used to start the fatigue tests because this is the yield stress of the substrate, which is ASTM A516 carbon steel and has this tensile strength at Grade 60. Thus, the highest level of fatigue stress was the yield strength. All the other levels were below the yield strength, in the elastic range of the steel, since this is the range of the several mechanical design where the phenomenon of fatigue fractures due to cyclic loading can occur.”

Question 7. English and format should be further checked and verified thoroughly. Page 8 Line 218, the phrase “you can observe the laminar architecture of the layer” should be revised (e.g., it can be observed).

Response: We agree with this recommendation. We changed the sentence as follows: “In Figure 7a it can be observed the laminar architecture of the layer, which has a splat morphology of different natures”. We also revised the whole manuscript to check another grammar and spelling issues.

Thanks again to the editor and reviewers for their contributions to improving the quality of the paper so that it can be published in Materials. We hope we have answer appropriately to all recommendations made by reviewers. Nonetheless, e remain at your disposal to resolve any further issue.

Respectfully submitted,

Rosivânia da Paixão Silva Oliveira

Universidade Federal do Rio Grande do Sul

Programa de Pós-graduação em Engenharia de Minas, Metalúrgica e de Materiais

Av. Paulo Gama, 110 – Farroupilha, Porto Alegre, RS – Brasil, CEP. 90040-060

Reviewer 2 Report

English grammatical mistakes and some typos must be corrected. Proper Scale bars in microscopy images must be placed. Manuscript quality can be improved by relating the results with proper discussion and citing literature. Results for fatigue testing should be elaborated in detail and citing the ASTM standards.

Author Response

Aracaju, January 30th 2023

Mr. Professor Reviewer 2

RE: Article submission

This is the revised version of the article entitled “Influence of grain size of Cr3C2-25 NiCr spray powder on the fatigue life of HVOF-sprayed coating on ASTM A516 steel substrate”. Thank you very much for allowing us to submit the revised manuscript. We would like to thank the your recommendations. We carefully analyzed all comments, as follows. All changes made into the manuscript are marked in yellow.

Reviewer 2.:

As per reviewer comment 2: English grammatical mistakes and some typos must be corrected. Proper Scale bars in microscopy images must be placed. Manuscript quality can be improved by relating the results with proper discussion and citing literature. Results for fatigue testing should be elaborated in detail and citing the ASTM standards.

Response: We would like to thank the reviwer for this recomendation. We have revised the grammar and typos issues. We have corrected the bars of the images. The quality of the manuscript were improved by linking the results with appropriate discussion and citing the literature. The fatigue results have been presented in more detail and the ASTM standard has been cited. All these changes are marked in yellow in the manuscript.

Thanks again to the editor and reviewers for their contributions to improving the quality of the paper so that it can be published in Materials. We hope we have answer appropriately to all recommendations made by reviewers. Nonetheless, e remain at your disposal to resolve any further issue.

Respectfully submitted,

Rosivânia da Paixão Silva Oliveira

Universidade Federal do Rio Grande do Sul

Programa de Pós-graduação em Engenharia de Minas, Metalúrgica e de Materiais

Av. Paulo Gama, 110 – Farroupilha, Porto Alegre, RS – Brasil, CEP. 90040-060

Reviewer 3 Report

The paper describes RT fatigue of A516 steel with HVOF sprayed composite coating formed by Cr3C2 particles in NiCr matrix (25wt%). Commercially available powder was subjected to high entropy milling (HEM) to obtain fine agglomerated powder. The same deposition parameters were used for both as-received and HEM powder. The thin (40-50um) coatings as well as the feedstock powders analysed by SEM,TEM,  and XRD to find finer crystallites in HEM feedstock and deposit and also finer splat size in the HEM deposit. The coated substrates were subjected to 3pb loading at R=0.1 with coating in tension and the obtained S-N curves were fitted and evaluated to conclude that HEM coating increases fatigue limit above that of bare substrte and coating from as-received powder. At the same time, small fatigue life decrease was observed for all coated specimen.

Paper covers interesting topic, and is systematically written in good English. Some minor details need to be corrected to facilitate understanding, therefore MINOR revision is suggested.

l28: protect substrates XXX damaging, missing preposition

l74: heat consumption -> heat transfer to feedstock ?

l76: HVOF higher temperature (??? HVOF is usually colder than plasma )

l105: featuring-> identification ?

l115-Kv->kV

l115 samples -> powders ?

l183 – rephrasing sudden -> fast, asymptotically approaching saturation, etc…

l186 – the amount of ground particles ????, there are agglomerated fragments of the original particles in the fig.

l188 -  MET -> TEM

fig 7, marking the carbides, Ni matrix and carbides, porosity

l246 –ssentence not clear, machining process should decrease roughness, thus it should improve fatigue, the sentence seems to say opposite

l280 and fig 8 – the discussion on carbides should be made more clear, please explain “thinner carbides” and “higher carbides” . Please mark carbides in fig 8..

l255: 550MPa is said to be yield strength, then the fatigue test at 550MPa are thus not in linear elastic range and below yield.

l317: isn’t the ‘better fatigue performance of milled coating’ too strong statement ? Milled coatings exhibited in fact lower fatigue life above fatigue limit and only the limit itself was higher.

Author Response

Aracaju, January 30th 2023

Mr. Professor Reviewer 3

RE: Article submission

This is the revised version of the article entitled “Influence of grain size of Cr3C2-25 NiCr spray powder on the fatigue life of HVOF-sprayed coating on ASTM A516 steel substrate”. Thank you very much for allowing us to submit the revised manuscript. We would like to thank the your recommendations. We carefully analyzed all comments, as follows. All changes made into the manuscript are marked in yellow.

Reviewer 3.:

Question 1. l28: protect substrates XXX damaging, missing preposition

Response: We changed the text accordingly: “Protective coating can increase the lifespan of equipment parts and structures, since it protects substrates of damaging mechanisms, such as corrosion, fatigue, high-temperature oxidation, and wear.”

Question 2. l74: heat consumption -> heat transfer to feedstock ?

Response: We agree with this recomendation. We changed the text as follows: “The HVOF hamper phase changes, due to the faster powder speed, heating and cooling and the lower heat transfer to feedstock. HVOF-sprayed nanostructured materials can provide coatings more resistant to crack propagation [14].”

Question 3. l76: HVOF higher temperature (??? HVOF is usually colder than plasma)

Response: We would like to thanks the reviewer for this recommendation. We changed the text as follows: “Although conventional thermal spraying makes particles melt, HVOF's temperature and spraying speed cause plastic strain in the powder upon impact against the substrate and form mechanically bonded flattened portions of the deposited material, sometimes called splats, that cool at rates up to 1x106 K/s”.

Question 4. l105: featuring-> identification ?

Response: Thank you for this comment. We changed the text accordingly: “X-ray diffraction (XRD) analysis enabled identification of the phases of the powder after milling.”

Question 5. l115-Kv->kV

Response: Thank you for this comment. We corrected it in the text.

Question 6. l115 samples -> powders ?

Response: Thank you for this comment. We changed the text accordingly: “Scanning electron microscopy (SEM TESCAN VEGA LMS) and energy dispersive spectroscopy (EDS JOEL Carry Scope JSM-5700) techniques were applied for powder and coating analysis purposes.”

Question 7. l183 – rephrasing sudden -> fast, asymptotically approaching saturation, etc…

Response: Thank you for this comment. We changed the text accordingly: “There was a fast decrease in the size before the 5-hour milling, asymptotically approaching saturation value of the size reduction.”

Question 8. l186 – the amount of ground particles ????, there are agglomerated fragments of the original particles in the fig.

Response: We would like to thanks the reviewer for this recommendation. We changed the text as follows: “The agglomeration of fragments of the original particles (Figure 4a and 4b) may occur after 12 hours of grinding due to their high surface energy and electrostatic strengths between particles.”

Question 9. l188 -  MET -> TEM

Response: Thank you for this comment. We changed the text.

Question 10. fig 7, marking the carbides, Ni matrix and carbides, porosity

Response: We would like to thank the rewiver for this recomendation and we changed the text, as follows: “Dark arrows denote carbides surrounded by the matrix. White arrows denote some porosities.”

Question 11. l246 – sentence not clear, machining process should decrease roughness, thus it should improve fatigue, the sentence seems to say opposite

Response: Thank you for this comment. We agree with this recomendation. We changed the manuscript as follows: “Roughness is significantly influenced by the type of coating. The coated surface of a given component is usually machined after the coating process to reduce the roughness of the coated surface. However, the machining process was not the subject of this study.

Question 12. l280 and fig 8 – the discussion on carbides should be made more clear, please explain “thinner carbides” and “higher carbides” . Please mark carbides in fig 8..

Response: We would like to thanks the reviewer for this recommendation. The terms “thinner carbides” and “higher carbides” were changed. We marked in fig 8 and we changed the text accordingly: “The fatigue limit reached 380 MPa for the Milled coating, which tends to be 40 MPa higher than for the other coatings (Figure 9). The development of fatigue cracks can be accelerated by increasing the roughness [19].”

Question 13. l255: 550MPa is said to be yield strength, then the fatigue test at 550MPa are thus not in linear elastic range and below yield.

Response: We agree with this recomendation and we changed the text accordingly: “The value of 550 MPa was used to start the fatigue tests because it is the yield stress of the substrate, which is ASTM A516 carbon steel and has this tensile strength at Grade 60.” More about this question regarding to elastic/plastic range was changed according to the Question 6 of Reviewer 1.

Question 14. l317: isn’t the ‘better fatigue performance of milled coating’ too strong statement ? Milled coatings exhibited in fact lower fatigue life above fatigue limit and only the limit itself was higher.

Response: We would like to thank the rewiver for this recomendation: “Milled coatings exhibited in fact lower fatigue life above fatigue limit and only the limit itself was higher.”

Thanks again to the editor and reviewers for their contributions to improving the quality of the paper so that it can be published in Materials. We hope we have answer appropriately to all recommendations made by reviewers. Nonetheless, e remain at your disposal to resolve any further issue.

Respectfully submitted,

Rosivânia da Paixão Silva Oliveira

Universidade Federal do Rio Grande do Sul

Programa de Pós-graduação em Engenharia de Minas, Metalúrgica e de Materiais

Av. Paulo Gama, 110 – Farroupilha, Porto Alegre, RS – Brasil, CEP. 90040-060

Reviewer 4 Report

I think this manuscript could be much better after additional tests, eg porosity measurements and fatigue studies of uncoated steel. Below are my detailed comments:

Materials and Methods

lines 135-138: What was the depth of indentations during hardness measurements? Add it, please.

Results and Discussion

Figure 2 shows the disappearance of the Cr7C3 phase with increasing the Cr2O3 phase and milling time. Why this was not mentioned?

The image in Figure 6 should have the same magnification as the images in Figure 5.

Does Figure 7 show the spray powders or the coatings? Lines 210-211 mention coatings, but there is powder in the figure's caption.

Table 2: Add what the Ra roughness parameter is shown in Table 2.
Add the porosity of the coatings

Lines 263-269: There are no results for the uncoated specimen, so such a statement that "the original coating contributed more to increasing the fatigue resistance of the specimen than the other coatings" cannot be written."

Fatigue resistance shows the stress at which the material does not break after 10^6 cycles. Thus, the number of cycles is used to determine the stress. It cannot be said that the coating contributed more to the fatigue strength of the sample, as it withstood more cycles to failure with less stress, as shown in Figure 9. It should be emphasized that in the case of fatigue, the measure of resistance is the stress at which the sample can withstand 10^6 cycles. The best fatigue resistance had a milled coating.

In the discussion of the fatigue test results the porosity should be included.

Show the nucleation sites more precisely. It is important whether they occurred on the surface of the coating, inside it or under the coating. The images in Figure 10 do not show this.

Author Response

Aracaju, January 30th 2023

Mr. Professor Reviewer 4

RE: Article submission

This is the revised version of the article entitled “Influence of grain size of Cr3C2-25 NiCr spray powder on the fatigue life of HVOF-sprayed coating on ASTM A516 steel substrate”. Thank you very much for allowing us to submit the revised manuscript. We would like to thank the your recommendations. We carefully analyzed all comments, as follows. All changes made into the manuscript are marked in yellow.

Reviewer 4.:

As per reviewer comment 4: I think this manuscript could be much better after additional tests, eg porosity measurements and fatigue studies of uncoated steel. Below are my detailed comments:

Response: Porosity in the three groups was less than 2%. Since the porosity was lower than the resolution of the technique used to measure it, we set the value below 2% for all groups. Moreover, in the literature, the porosity lower than 2% is considered appropriate to the deposition technique used. The objective of the article was to compare the fatigue performance of the three groups in order to verify the effect of particle refinement on fatigue, which is why we do not present the curve of the base metal (substrate).

Question 1. lines 135-138: What was the depth of indentations during hardness measurements? Add it, please.

Response: We would like to thank the rewiever for this recomendation. We think that the most important dimension to be informed would be the smallest diagonal of the diamond indentation  of microhardness (Knoop), since we measure the microhardness in the transverse plane of the coating layer. Therefore, the diagonal of the indentation must be smaller than the coating thickness. The value of this diagonal was added to the manuscript.

Results and Discussion

Question 2. Figure 2 shows the disappearance of the Cr7C3 phase with increasing the Cr2O3 phase and milling time. Why this was not mentioned?

Response: We would like to thank the rewiver for this recommendation. We added the following sentence to the paper: “Figure 2 also denotes the disappearance of the Cr7C3 phase with increasing the Cr2O3 phase and milling time.”

Question 3. The image in Figure 6 should have the same magnification as the images in Figure 5.

Response: Thank you for this comment. We changed the figure to the same magnification as the other images.

Question 4. Does Figure 7 show the spray powders or the coatings? Lines 210-211 mention coatings, but there is powder in the figure's caption.

Response: Thank you for this comment. We agree with this recommendation. We changed the text as follows: “Figure 7. SEM images of coating with powder (a) milled, (b) original and (c) 50% mixed.”

Question 5. Table 2: Add what the Ra roughness parameter is shown in Table 2. Add the porosity of the coatings

Response: Thank you for this comment. We added the Ra parameter, but the porosity of the coatings is the same for all coatings.

Question 6. Lines 263-269: There are no results for the uncoated specimen, so such a statement that "the original coating contributed more to increasing the fatigue resistance of the specimen than the other coatings" cannot be written."

Response: Thank you for this comment. This sentence was removed from the text.

Question 7. Fatigue resistance shows the stress at which the material does not break after 10^6 cycles. Thus, the number of cycles is used to determine the stress. It cannot be said that the coating contributed more to the fatigue strength of the sample, as it withstood more cycles to failure with less stress, as shown in Figure 9. It should be emphasized that in the case of fatigue, the measure of resistance is the stress at which the sample can withstand 10^6 cycles. The best fatigue resistance had a milled coating.

Response: Thank you for this comment. We changed the text accordingly: “It should be emphasized that in the case of fatigue, the measure of resistance is the stress at which the sample can withstand 10^6 cycles. The best fatigue resistance had a milled coating.”

Question 8. In the discussion of the fatigue test results the porosity should be included.

Response: Thank you for this comment. The porosity of the coatings did not show significant differences for all types of coatings, so it was not a study parameter.

Question 9. Show the nucleation sites more precisely. It is important whether they occurred on the surface of the coating, inside it or under the coating. The images in Figure 10 do not show this.

Response: Thank you for this comment. We added an image to highlighting the surface of the coating as the fatigue nucleation site.

Thanks again to the editor and reviewers for their contributions to improving the quality of the paper so that it can be published in Materials. We hope we have answer appropriately to all recommendations made by reviewers. Nonetheless, e remain at your disposal to resolve any further issue.

Respectfully submitted,

Rosivânia da Paixão Silva Oliveira

Universidade Federal do Rio Grande do Sul

Programa de Pós-graduação em Engenharia de Minas, Metalúrgica e de Materiais

Av. Paulo Gama, 110 – Farroupilha, Porto Alegre, RS – Brasil, CEP. 90040-060

Round 2

Reviewer 1 Report

The author has addressed all the comments in the revised version. This version clearly presents the effect of grain size of powders on the fatigue life of Cr3C2-25NiCr coatings, which is of interest to scientific community. The results in this work are of importance to the application of the high-energy milling method in producing high fatigue life of Cr3C2-25NiCr coatings, and are useful as a basis for comparison for more in-depth future works. English and grammatical mistakes must be checked and corrected. In general, the revised version needs minor revisions before its publication.

Author Response

Aracaju, February 05th 2023

Mr. Professor Octavian Barbos

Special Issue Editor, MDPI Romania

Materials Editorial Office

RE: Article submission

Dear Professor Octavian Barbos

This is the revised version of the article entitled “Influence of grain size of Cr3C2-25 NiCr spray powder on the fatigue life of HVOF-sprayed coating on ASTM A516 steel substrate”. Thank you very much for allowing us to submit the revised manuscript again. We would like to thank the reviewers’ recommendations. We carefully analyzed all reviewers' comments, as follows. All changes made into the manuscript are marked in yellow.

Reviewer 1.:

As per reviewer comment 1: “The author has addressed all the comments in the revised version. This version clearly presents the effect of grain size of powders on the fatigue life of Cr3C2-25NiCr coatings, which is of interest to scientific community. The results in this work are of importance to the application of the high-energy milling method in producing high fatigue life of Cr3C2-25NiCr coatings, and are useful as a basis for comparison for more in-depth future works. English and grammatical mistakes must be checked and corrected. In general, the revised version needs minor revisions before its publication..”

Response: Thank you for these comments. English and grammatical mistakes were checked and corrected.

Finally, we replaced the term "grain" with "powder" in the title, as the latter is more appropriate for the present study.

Thanks again to the editor and reviewers for their contributions to improving the quality of the paper so that it can be published in Materials. We hope we have answer appropriately to all recommendations made by reviewers. Nonetheless, we remain at your disposal to answer of any further issue.

Respectfully submitted,

Rosivânia da Paixão Silva Oliveira

Universidade Federal do Rio Grande do Sul

Programa de Pós-graduação em Engenharia de Minas, Metalúrgica e de Materiais

Av. Paulo Gama, 110 – Farroupilha, Porto Alegre, RS – Brasil, CEP. 90040-060

Reviewer 4 Report

The authors did not take into account some of my earlier comments, so I still have them:

lines 135-139: add the depth of indentations; It is very important.

Figures 5 and 6 should have the same magnification.

Most of the images are very dark, so they show almost nothing. They should be brighter.

Show the nucleation sites more precisely. It is important whether they occurred on the surface of the coating, inside it or under the coating. The images in Figure 10 do not show this.

Author Response

Aracaju, February 05th 2023

Mr. Professor Octavian Barbos

Special Issue Editor, MDPI Romania

Materials Editorial Office

RE: Article submission

Dear Professor Octavian Barbos

This is the revised version of the article entitled “Influence of grain size of Cr3C2-25 NiCr spray powder on the fatigue life of HVOF-sprayed coating on ASTM A516 steel substrate”. Thank you very much for allowing us to submit the revised manuscript again. We would like to thank the reviewers’ recommendations. We carefully analyzed all reviewers' comments, as follows. All changes made into the manuscript are marked in yellow.

Reviewer 4.:

Question 1. lines 135-139: add the depth of indentations; It is very important.

Response: We would like to thank the reviewer for this recomendation. We added the following sentence to the paper: The maximum depth of indentation of the coatings was 0.77 microns.

Question 2. Figures 5 and 6 should have the same magnification.

Response: Thank you for this contribution. We changed the figures properly.

Question 3. Most of the images are very dark, so they show almost nothing. They should be brighter.

Response: Thank you for this recommendation. We improved the brightness of the images.

Question 4. Show the nucleation sites more precisely. It is important whether they occurred on the surface of the coating, inside it or under the coating. The images in Figure 10 do not show this.

Response: We would like to thank the reviewer for this recommendation. We have made more evident the place of fatigue nucleation on the surface of the coating. We also added the following sentence to the paper: Fatigue nucleated on the surface of the coatings, as exhibited in Figure 11, which denotes that there are fatigue striations on the coating coming from the surface. The limit between the fracture plane and the coating surface is indicated in Figure 11b.

Finally, we replaced the term "grain" with "powder" in the title, as the latter is more appropriate for the present study.

Thanks again to the editor and reviewers for their contributions to improving the quality of the paper so that it can be published in Materials. We hope we have answer appropriately to all recommendations made by reviewers. Nonetheless, we remain at your disposal to answer of any further issue.

Respectfully submitted,

Rosivânia da Paixão Silva Oliveira

Universidade Federal do Rio Grande do Sul

Programa de Pós-graduação em Engenharia de Minas, Metalúrgica e de Materiais

Av. Paulo Gama, 110 – Farroupilha, Porto Alegre, RS – Brasil, CEP. 90040-060
